# The decline of the 2022 Italian mpox epidemic: Role of behavior changes and control strategies

Giorgio Guzzetta [1], Valentina Marziano [1], Alessia Mammone[2], Andrea Siddu[2], Federica Ferraro[2], Anna Caraglia[2], Francesco Maraglino[2], Giovanni Rezza[2,3], Alessandro Vespignani [4], Ira Longini[5], Marco Ajelli [6] & Stefano Merler [1] ✉

In 2022, a global outbreak of mpox occurred, predominantly impacting men who have sex with men (MSM). The rapid decline of this epidemic is yet to be fully understood. We investigated the Italian outbreak by means of an individual-based mathematical model calibrated to surveillance data. The model accounts for transmission within the MSM sexual contact network, in recreational and sex clubs attended by MSM, and in households. We indicate a strong spontaneous reduction in sexual transmission (61-87%) in affected MSM communities as the possible driving factor for the rapid decline in cases. The MSM sexual contact network was the main responsible for transmission (about 80%), with clubs and households contributing residually. Contact tracing prevented about half of the potential cases, and a higher success rate in tracing contacts could significantly amplify its effectiveness. Notably, immunizing the 23% of MSM with the highest sexual activity (10 or more partners per year) could completely prevent new mpox resurgences. This research underscores the importance of augmenting contact tracing, targeted immunization campaigns of high-risk groups, and fostering reactive behavioral changes as key strategies to manage and prevent the spread of emerging sexually transmitted pathogens like mpox within the MSM community.

A large epidemic of mpox disease (formerly known as monkeypox[1]) caused by Clade IIb (formerly West African clade[2]) of the monkeypox virus (MPV) has spread globally since early May 2022, with a worldwide total of 89,391 lab-confirmed cases and 153 deaths by August 16, 2023[3,4]. This epidemic represents the first case of widespread transmission outside of Africa where self-contained outbreaks of zoonotic origin dominated by intra-familiar transmission had been occurring for decades with increasing frequency[5,6]. Contact with body fluids and affected skin appears to be the most important route of transmission,

but other routes, such as respiratory droplet transmission via intimate contact, cannot be excluded[6]. The 2022 global outbreak was largely driven by sexual transmission among men who have sex with men (MSM)[2]. It is still unclear whether the different epidemiology of this outbreak depends on genomic adaptations of the circulating virus[7,8] or on its spread within a subpopulation at substantially higher risk. The initial cases of local transmission during the 2022 global mpox outbreak surfaced in the United Kingdom in early May. By July 23, the outbreak had broadened its reach to 75 countries, prompting the

[1]Center for Health Emergencies, Bruno Kessler Foundation, Trento, Italy. [2]Health Prevention Directorate, Ministry of Health, Rome, Italy. [3]Vita-Salute San Raffaele University, Milan, Italy. [4]Laboratory for the Modeling of Biological and Socio-Technical Systems, Northeastern University, Boston, MA, USA. [5]Department of Biostatistics, Colleges of Public Health and Health Professions, and Medicine, University of Florida, Gainesville, FL, USA. [6]Laboratory for Computational Epidemiology and Public Health, Department of Epidemiology and Biostatistics, Indiana University School of Public Health, Bloomington, IN, USA. ✉e-mail: merler@fbk.eu

World Health Organization to declare the mpox epidemic a Public Health Emergency of International Concern (PHEIC)[9]. By that time, Europe had been the most affected region, accounting for 72% of mpox cases reported globally. In several European nations, the early cases demonstrated associations with international travel, attendance at parties, MSM-specific recreational venues (e.g., saunas, sex clubs), and pride parades. This suggests that international amplification events played a crucial role in disseminating the infection across multiple locations[10]. A rapid expansion of reported cases occurred until the second week of July, when the incidence in Europe plateaued at about 2,500 cases per week until mid-August. In Italy the first case was confirmed in Rome, Italy, on May 19, 2022, and as of August 5th, Italy recorded more that 500 cases. Subsequently, there was a rapid downward trajectory in case numbers[11]. Since November 2022, less than 100 cases per week have been reported in the whole continent[11]. A Live Modified Vaccinia Virus Ankara (commercialized as Jynneos in the United States and as Imvanex in Europe), previously developed against smallpox and highly effective against mpox infection[12] was approved for use in Europe since early July 2022, but has not been massively deployed until the fall of the same year due to the unavailability of doses. Different reasons may explain the decline of viral circulation, such as the buildup of population immunity in groups at the highest sexual activity[13–16], a spontaneous behavior change in MSM communities affected by viral circulation[15,17], the impact of control interventions by public health authorities, or a combination of them. Here, we utilize an individual-based model calibrated with data from the Italian outbreak. Our aim is to examine various hypotheses, estimate the efficacy of implemented interventions along with potential alternative strategies, and determine immunization rates to preempt future outbreaks.

## Results

We implemented an individual-based model considering infectious contacts among MSM and between MSM and their household members as main routes of transmission, through a simulation period going from May 9, 2022, to February 28, 2023. Transmission among MSM occurred over a network of sexual partnerships where individuals were assigned a number of yearly sexual partners according to their highly heterogeneous empirical distribution[18] (see Table 1). High-risk (HR) individuals (here defined as those with 30 or more yearly sexual partners) were assumed to attend recreational clubs for MSM, where transmission among attendees who were not already paired within the sexual network was also possible. We considered a possible reduction of transmissibility in the sexual network and in clubs starting from June 8, to reproduce an analogous reduction in the net reproduction number observed in Italy for the same period[10]. Self-reporting of cases (infected individuals that are diagnosed after health-seeking), tracing of their sexual and household contacts, and the vaccination program (which started on August 8 in Italy) were implemented in the model (see Methods and Supplementary Material). We assume that 20% of sexual contacts for self-reported cases were effectively traced, and subject this assumption to validation after model calibration. Free model parameters were calibrated against the observed total number of cases with symptom onset in four epidemiological periods (Period 1: May 9–June 7; Period 2: June 8–July 7; Period 3: July 8–August 6; Period 4: August 7–September 5) and against a classification of a subset of

cases in the first two periods, obtained from epidemiological investigations (Fig. 1 and see Methods and Supplementary Material for the calibration details). The model reproduces well the daily variability in the number of diagnosed cases by date of symptom onset and the proportion of cases that are diagnosed via contact tracing, which were not provided during the calibration phase (Fig. 1). A median of 231 (90% prediction interval, PI: 9-5,983) cases were estimated to have been missed by the surveillance system.

The posterior distribution of the self-reporting ratio had a mean of 78% and a broad 95% credible interval (CrI) of 27-100% (Fig. 2). A massive reduction in transmissibility, both in clubs (mean 87%, 95%CI: 47-100%) and in the general sexual network (mean 61%, 95%CI: 27-87%) is necessary for the model to reproduce observed trends in data (Fig. 2).

The model suggests that, during the first period, 79% (95% Prediction Intervals, PI: 63–90%) of all infections were acquired through the sexual network, 11% (95%PI: 2–27%) in clubs, and 10% (95%PI: 4–18%) in households (Fig. 3). Due to the estimated reduction of transmissibility, the share of infections acquired through the sexual network in Period 4 decreased to 72% (95%PI: 55–88%), the share of those acquired in clubs decreased to 3% (95%PI: 0–12%), while the one relative to household transmission increased to 25% (95%PI: 11–39%). During the initial period, the average reproduction number is estimated to be 1.97 (95%PI: 1.50–2.51), but this number experienced a sharp decline beginning in the second period, reaching a value of 0.87 (95%PI: 0.74–0.97) in the fourth period (see Fig. 3). The reproduction number specific to the sexual contact network alone is estimated at 1.52 (95%PI: 0.97–2.23) during the first period - a value sufficient to initiate an outbreak even in the absence of other transmission routes. However, following the significant behavioral change, the reproduction number within the sexual contact network dropped to 0.63 (95% PI: 0.49–0.78). In the initial period, the estimated average number of secondary cases per infectious individual in club settings is at the high value of 2.75, though with a wide range of uncertainty (95%PI: 0.54–6.17). However, given that the model allowed club attendance only once a week and restricted it to a subset of individuals with very high-risk profiles, clubs' overall contribution to transmission remained relatively minor. Following the decrease in club transmissibility, the reproduction number linked to clubs dropped to 0.20 (95%PI: 0.00–0.87) by the fourth period. Notably, the estimated reproduction number associated with household transmissions consistently remained below the threshold, suggesting the occurrence of sporadic infections among household members.

According to our model, the depletion of susceptible individuals was not the primary driver for the epidemic's decline, even within groups characterized by high sexual activity. By the end of the fourth period, we estimate the cumulative incidence rate among high-risk individuals at 2.4 per 1000 (95%PI: 0.3-15.7 per 1000). This rate is markedly lower among individuals with fewer sexual partners (see Fig. 3). Table 1 reports a breakdown of infection proportions across different sexual activity groups.

We evaluated potential alternative public health strategies for this outbreak with respect to the implemented interventions in the baseline scenario. Our findings indicate that without contact tracing, the number of infections during the simulation period (from May 9, 2022, to February 28, 2023) could have doubled (see Fig. 4). Other mitigation

**Table 1 | Sexual activity groups and their share of the population and of all infections**

| Sexual activity group | Yearly sexual partners | Population share | Estimated share of all infections |
|---|---|---|---|
| Very low risk (VLR) | 1–3 | 47.9% | 9.2% (95%PI: 6.4-13.6%) |
| Low risk (LR) | 4–10 | 29.1% | 27.3% (95%PI: 22.9-32.7%) |
| Moderate risk (MR) | 11–29 | 20.5% | 46.8% (95%PI: 39.7-53.4%) |
| High risk (HR) | 30+ | 2.5% | 16.5% (95%PI: 8.5-27.1%) |

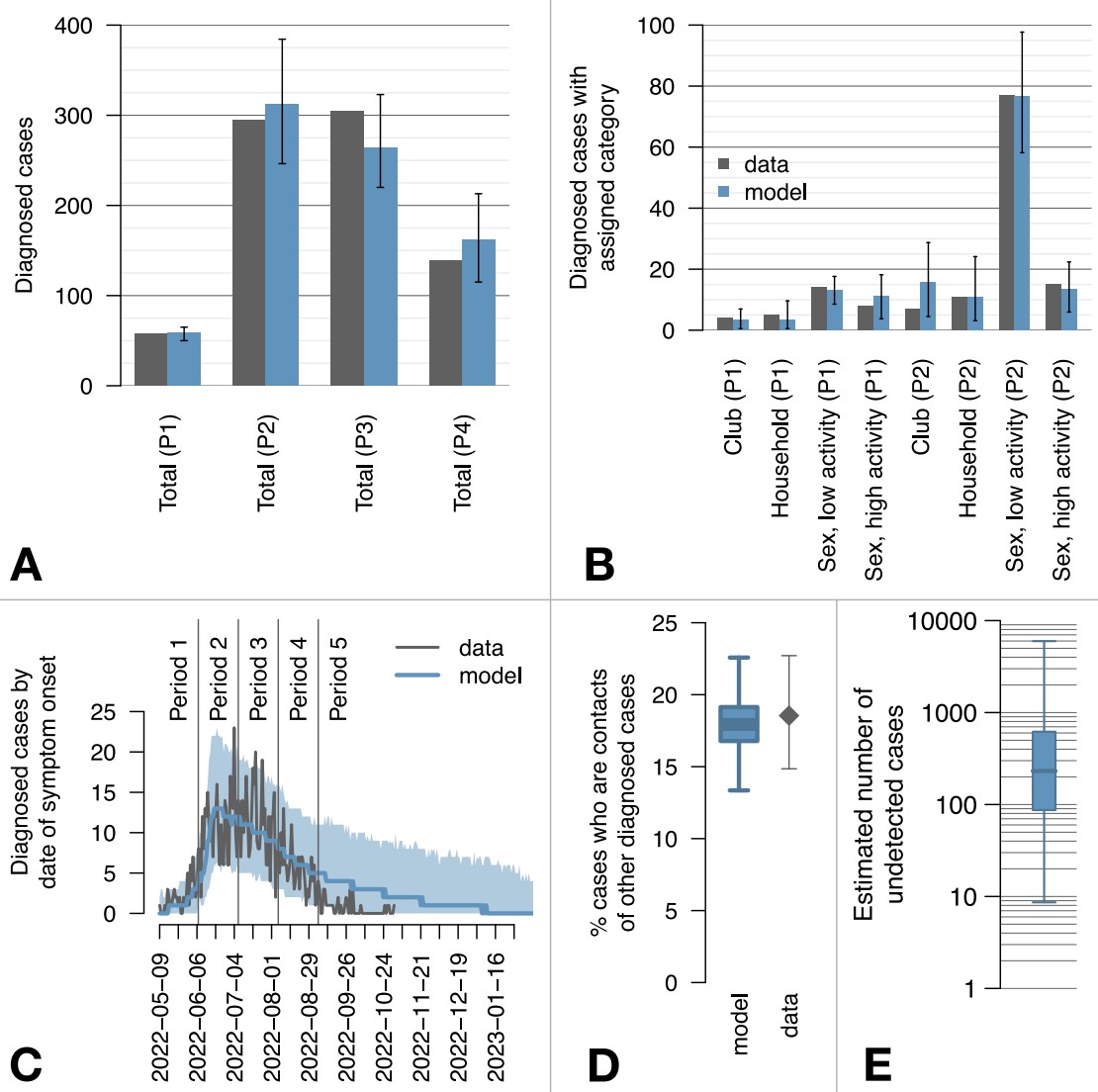

**Fig. 1 | Modeled and observed epidemiological dynamics during the mpox outbreak in Italy, 2022. A** Total number of diagnosed cases with symptom onset in four epidemiological periods (P1: May 9–June 7, 2022; P2: June 8–July 7; P3: July 8–August 6; P4: August 7–September 5). Bars represent the observed values (dark gray) and the mean modelled estimates (blue); vertical bars represent 95% prediction intervals (PI). **B** Classification of cases in periods P1 and P2 for the subset of cases for which a categorization was available. Categories represent the potential setting of exposure, with "low activity" and "high activity" representing individuals with 2 or less and 3 or more sexual partners disclosed in the preceding 3 weeks, respectively (see Methods). Bars represent the observed values (dark gray) and the mean modelled estimates (blue); vertical bars represent 95% PI. **C** Epidemic curve by symptom onset date. Observed data (dark gray line) are compared against the mean modeled curve (blue line) and the 95% PI. **D** Percentage of cases diagnosed via contact tracing; the boxplot represents the mean (central bar), interquartile range (IQR, rectangular box) and 95% PI (whiskers); the gray diamond and whiskers represent the mean and 95% confidence interval (CI) of the binomial distribution for the probability that a diagnosed case is a contact of another case in observed data. **E** Number of undetected mpox cases estimated by the model (y-axis in a log scale); central bar: median; box: interquartile range (IQR); whiskers: 90% PI. In all figures, model variability derives from 694 simulations accepted during calibration.

strategies, such as ring tracing (RT, i.e., tracing contacts of contacts), vaccination of contacts (CV), and ring vaccination (RV, i.e., vaccination of contacts of contacts), would not have substantially reduced the number of infections compared to the baseline scenario. We also show that the probability of spontaneous extinction of transmission chains due to stochasticity was low. However, the introduction of contact tracing significantly increased the likelihood of transmission chain extinction as time progressed (see Fig. 4).

Maintaining the baseline scenario of interventions while increasing the coverage of contact tracing (i.e., the proportion of successfully traced sexual contacts) to 40% could have resulted in an average reduction of 42% in the total number of infections compared to the baseline scenario (where coverage is at 20%). By increasing contact tracing coverage to 80%, we estimate an average reduction of 62% in

infections (Fig. 5). The probability of containing the outbreak at the source (i.e., inducing the extinction within the first month) increases from 10% in the baseline case to 16%, 27% and 45% with tracing coverage of 40%, 60% and 80%, respectively. Furthermore, the probability of disrupting all transmission chains increases earlier and more rapidly with increasing tracing coverage.

Lastly, we considered the potential impact against a possible future resurgence of mpox of a targeted vaccination campaign performed in addition to baseline interventions (contact tracing with 20% coverage) and aimed at groups with the highest sexual activity levels. Our findings suggest that to bring the effective reproduction number below the epidemic threshold (and therefore have close to 100% probability of outbreak control within one month) it would be sufficient to vaccinate all MR and HR individuals (i.e., all those with more

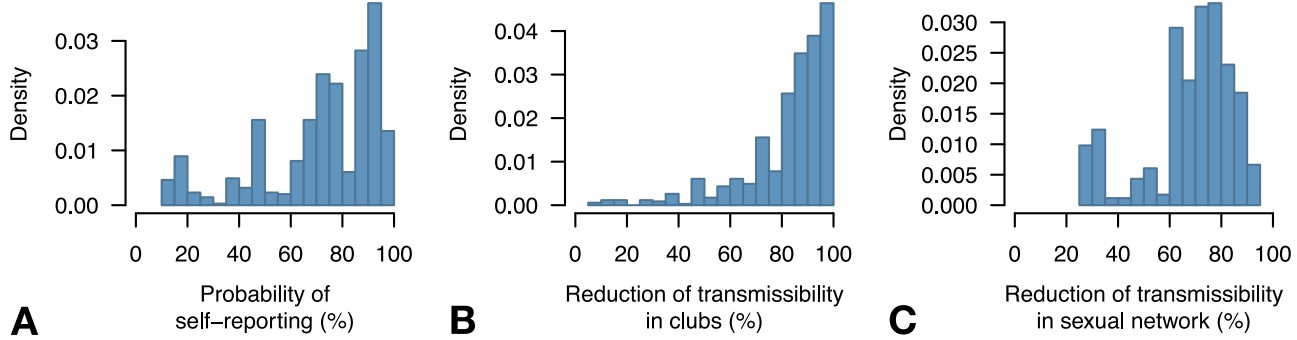

**Fig. 2 | Posterior distributions of parameters of public health interest.**
**A** Proportion of mpox infections that self-report symptoms to the surveillance system and are confirmed as mpox cases. **B** Reduction of the transmission rate in clubs due to spontaneous behavior change, assumed to be in place since June 8, 2022. **C** Reduction of the transmission rate in the sexual contact networks due to spontaneous behavior change, assumed to be in place since June 8, 2022.

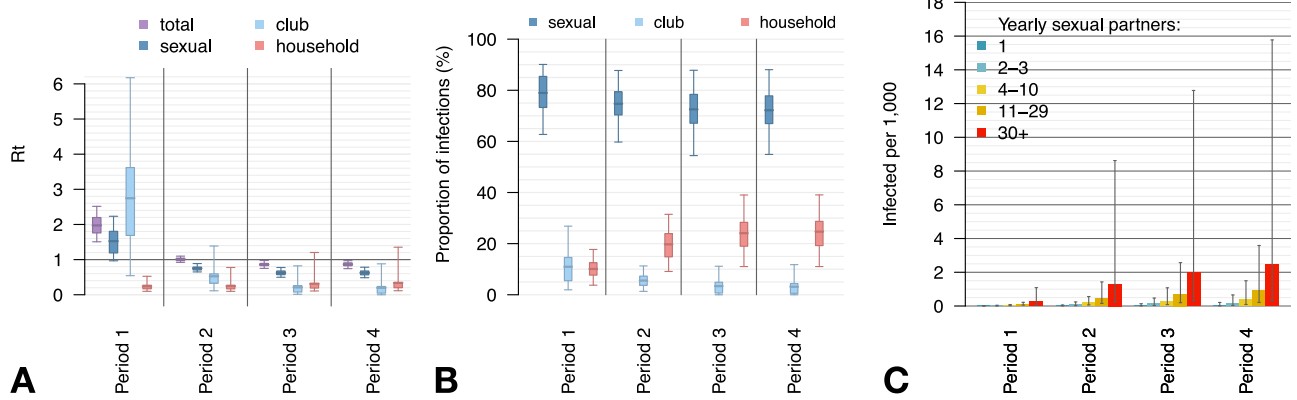

**Fig. 3 | Model insights on the transmission dynamics of the mpox outbreak in Italy, 2022. A** Reproduction number by route of acquisition and period (Period 1: May 9 – June 7, 2022; Period 2: June 8 – July 7; Period 3: July 8 – August 6; Period 4: August 7 – September 5). Boxplots represent mean values (central bars), inter-quartile ranges (rectangular boxes), and 95%PI (whiskers). **B** Proportion of infections by route of acquisition and period. Boxplots represent mean values (central bars), interquartile ranges (rectangular boxes), and 95%PI (whiskers). **C** Cumulative incidence rate by number of yearly sexual partners and period. Bars represent mean values and whiskers represent 95%PI.

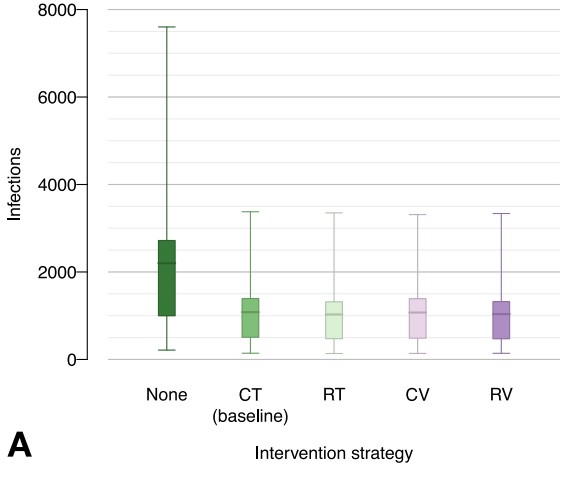
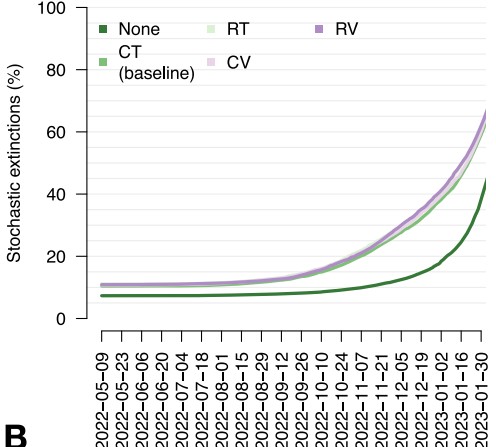

**Fig. 4 | Alternative intervention scenarios. A** Total number of infections across the simulation period (May 9, 2022 – February 28, 2023), by intervention. None: no active intervention; CT: contact tracing (tracing contacts of diagnosed cases); RT: ring tracing (tracing both contacts of diagnosed cases and contacts of contacts); CV: contact vaccination (tracing and vaccinating contacts of diagnosed cases); RV: ring vaccination (tracing and vaccinating contacts of diagnosed cases and contacts of contacts). Boxplots represent mean values (central bars), interquartile ranges (rectangular boxes), and 95%PI (whiskers). **B** Cumulative percentage of simulations that result in a stochastic extinction over time, by type of intervention. A stochastic extinction occurs when there are no more infectious or exposed individuals in the population.

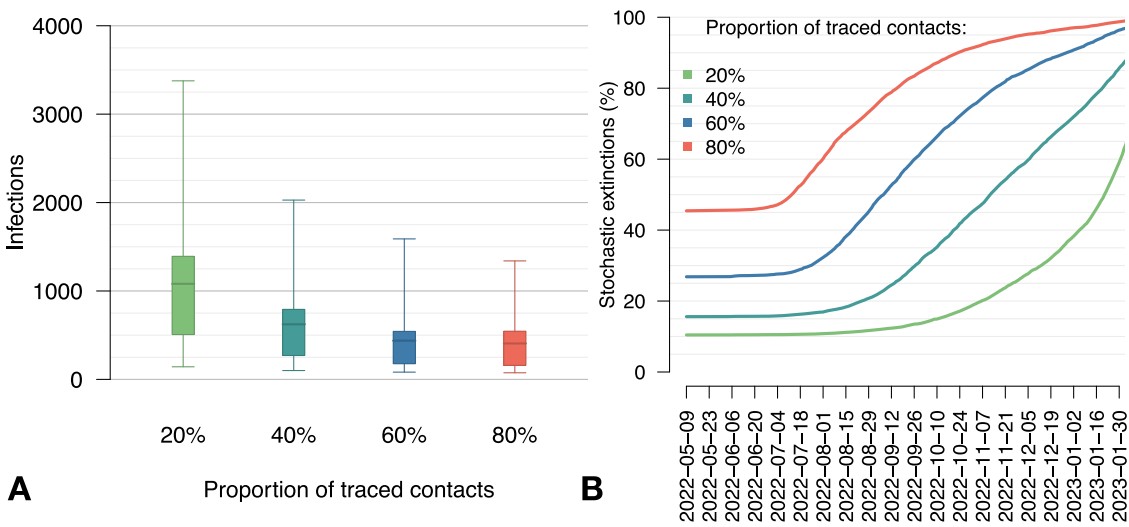

**Fig. 5 | Alternative proportions of sexual contacts that are successfully traced.**
**A** Total number of infections across the simulation period (May 9, 2022 – February 28, 2023), by proportion of traced contacts. Boxplots represent mean values (central bars), interquartile ranges (rectangular boxes), and 95%PI (whiskers).

**B** Cumulative percentage of simulations that result in a stochastic extinction over time, by proportion of traced contacts. A stochastic extinction occurs when there are no more infectious or exposed individuals in the population.

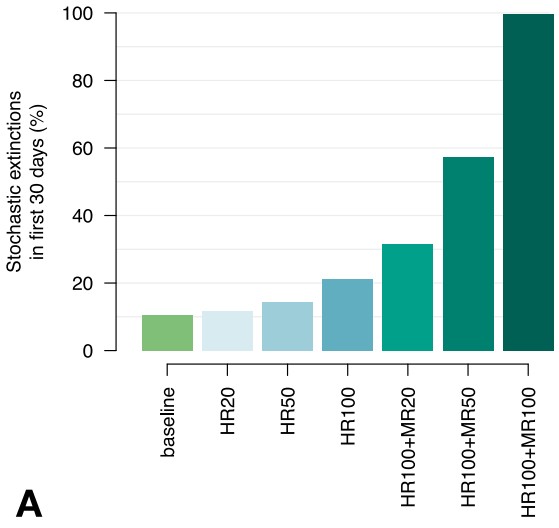
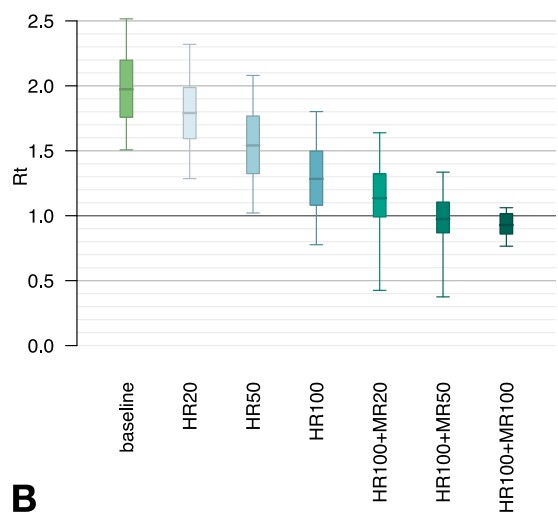

**Fig. 6 | Effect of preventive vaccination on a potential new mpox outbreak.**
**A** Percentage of simulations that end up in a stochastic extinction within 30 days since the first case, by vaccination coverage. A stochastic extinction occurs when there are no more infectious or exposed individuals in the population. HR20: vaccinating 20% of high-risk (HR) MSM, i.e., those with 30 sexual partners per year or more; HR50: vaccinating 50% of HR MSM; HR100: vaccinating all HR MSM;

MR20: vaccinating 20% of moderate risk (MR) MSM, i.e., those with 11 to 29 sexual partners per year; MR50: vaccinating 50% of MR MSM; MR100: vaccinating all MR MSM. **B** Reproduction number by vaccination coverage. Boxplots represent mean values (central bars), interquartile ranges (rectangular boxes), and 95%PI (whiskers).

than 10 sexual contacts per year, amounting to about 23% of the MSM population) (Table 1 and Fig. 6). Vaccinating all HR MSM and 50% of MR, corresponding to about 13% of the MSM population, would bring the mean effective reproduction number to about 1 and could prevent a new outbreak in 57% of cases.

In the Supplementary materials, we report four sensitivity analyses where we re-calibrated the model under different assumptions: (1) mixing in the sexual contact network is more assortative than the baseline with respect to sexual activity levels, i.e. MSM are more likely to encounter individuals with similar numbers of yearly sexual partners[19]; (2) mixing is both more assortative by sexual activity levels and assortative by age; (3) there are no upper bounds on the number of yearly partners in the sexual network; 4) individuals are allowed to

attend clubs independently of their sexual activity level. All results and conclusions remained substantially unchanged both qualitatively and quantitatively.

## Discussion

In our modeling study we estimated that spontaneous behavior change in MSM communities reduced the mpox transmission rate in the sexual network and in clubs by approximately 60 and 85%, respectively, at the beginning of June 2022. This reduction may have been induced both by a reduction in the frequency of sexual encounters and by an increase in safer sex practices and was possibly fostered by the spreading news about the outbreak both in the media and by word of mouth in the MSM population. The reactive behavior

change may refer to spatiotemporally local subpopulations where infection was known to circulate, rather than to a generalized and sustained one. Such a large reduction of transmission may have resulted in the interruption of local mpox transmission chains, such that even though normal sexual activity may have resumed in a given local community, it did not contribute to a resurgence of the epidemic. The mean probability of self-reporting for mpox infection was estimated to be 78%, but probabilities as low as 27% were also compatible with the observed data. This raises the question of whether a substantial, under-detected viral circulation may have induced saturation effects in high-risk groups, contributing to the outbreak fadeout. Model results suggest that this is not the case. The maximum cumulative attack rates of infection in the highest-risk population never exceeded 1.5%, even when considering the lowest admitted reporting rates.

This study also suggested that, in absence of behavior change, about 80% of infections were attributable to the sexual contact network, and transmission within households and clubs accounted for only about 10% respectively. Despite clubs holding the potential to amplify transmission, given their potentially higher reproduction number (average 2.7, compared to 1.5 in the sexual contact network), their role in spreading infections remained minor due to the relatively small size of the population frequenting them.

In our evaluation of additional intervention strategies, the vaccination of contacts and/or the tracing of secondary contacts (i.e., contacts of contacts) would have averted only a small number of infections. This is explained by the observation that almost two thirds of cases in the model are in MSM with more than 10 sexual partners per year. However, their contacts are likely to have a significantly smaller number of sexual partners (about half of MSM in Italy have 3 partners per year or less), contributing fewer contacts to the overall pool of traced individuals. Despite this feature of mpox transmission, ring vaccination does give active protection to those most exposed to cases of mpox, thus, providing individual level protection. Furthermore, we estimated the coverage of a targeted vaccination program that would be required in order to prevent a new mpox outbreak with a similar transmissibility and contact tracing interventions. We showed that vaccinating all high- and moderate-risk individuals (23% of the MSM population, about 380,000 individuals) could fully avoid a resurgence of the outbreak. Vaccinating all the high-risk population and 50% of the moderate-risk population (i.e., 13% of the MSM population; about 210,000 individuals) could avoid new outbreaks with probability greater than 50%. Vaccination initiatives play a crucial role in preventing a probable global recurrence of mpox[20].

Our results differ significantly from other published studies, where the accrual of immunity in highest risk groups was suggested as the sole responsible for the decline of the epidemics[13,14] or where behavioral change was found to have a lesser role in reducing mpox transmissibility, with an estimated reduction around 40–50%, compared to our 60-85% estimate[15,17]. These differences may be due to assumptions in the model (e.g., not considering the effect of contact tracing and behavior change[13,14]) and/or to the different epidemiological context. Both studies estimating a 40-50% reduction of transmissibility due to behavior change were focused on the UK[15,17], a country that experienced an almost 4-fold cumulative incidence of diagnosed cases compared to Italy[11], and where the sexual network is characterized by much heavier tails than those estimated from available data for Italy[13,18]. Furthermore, the much shorter average serial interval identified in contact tracing data in the UK compared to Italy (8–9 days vs. 12.5 days)[10,17,21] seems to suggest a more rapid identification of cases via contact tracing, so that a lower contribution of behavior change in the reduction of transmissibility may have been sufficient to downturn the epidemic.

Several limitations need to be considered when interpreting our results. The model does not consider asymptomatic cases and pre-symptomatic transmission. Asymptomatic cases are expected to be uncommon[22] and, if they can transmit, they are implicitly considered in the proportion of undiagnosed cases. Presymptomatic transmission seems possible[21], but its frequency has not been quantified accurately; if it is a significant feature of mpox transmission, our model may overestimate the effectiveness of contact tracing. However, the distribution of the simulated generation time for diagnosed cases in the model (mean 14.4 days) closely approximated the one previously estimated from contact tracing data for Italian cases (mean 12.5 days[10], see Supplementary material), suggesting that neglecting presymptomatic transmission does not impact on the ability of the model to correctly capture the timing of transmission.

There are many unknowns about the sexual contact network of MSM in Italy. This includes the proportion of MSM in the population and the aggregation of MSM in households. Here, we assumed that 10% of males between the ages of 15 and 70 are MSM, based on 2021 data from the United Kingdom Office for National Statistics[23], where 10% of men between the age of 16 and 34 did not identify as heterosexual. Although this percentage was lower (i.e., 7%) in age group 35–64, it is possible that some of the individuals who identify as heterosexual would engage in homosexual activity without disclosing it, due to persisting stigma; furthermore, we expect the high-risk sexual activity groups to be concentrated in the younger ages. Notably, these percentages have steadily increased each year since the first survey in 2014 where 6% of 16–34 years old and 5% of 35–64 did not identify as heterosexual[23]. These figures show a rapidly evolving scenario of sexual behaviors and/or assertiveness with respect to one's own sexual identity. Finally, in absence of better information, MSM individuals were distributed homogeneously across households, therefore neglecting the possible aggregation in households of homosexual couples, families, or groups of friends/roommates. However, given the limited impact of household transmission identified in this study, we expect this limitation to not affect the main conclusions. Another important source of uncertainty is the possible effect of geographical and temporal heterogeneities on sexual behavior. Unsurprisingly, over half of mpox cases in Italy were in individuals residing in four larger cities (Milan 29%, Rome 14%, Naples 5% and Bologna 5%); due to the lack of data, we could not take into account potential geographical differences in sexual behavior between different areas of the country. In addition, many of the initially detected cases were related to sexual activity during national or international travels, also associated with attendance to gay pride parties during May and June. It is possible that sexual activity may be associated with temporal patterns, possibly biasing the amount of the spontaneous reduction in risky behaviour estimated by the model.

It is worth remarking that despite the above limitations, the model was able to reproduce the daily variability in the number of diagnosed cases and the proportion of diagnosed cases that were a known contact of a case. The high estimated mean proportion of self-reported cases (78%) seems compatible with an infection that in most cases causes overt disease requiring medical attention, such as mpox[22]. The model also estimated an effective reproduction number of 1.97 (95%PI: 1.50–2.51) in the first phase, before a rapid decline of transmissibility to an effective reproduction number of 0.87 (95%PI: 0.74-0.97) later on in the epidemic. These numbers are compatible with a previous estimate of 2.43 (95%CI: 1.82-3.26) at the beginning of the Italian outbreak[10].

The findings of our study, derived from an individual-based model that accounted for transmission within households, clubs, and the associated sexual contact network, suggest that the decline of the mpox outbreak in Italy can be attributed to a spontaneous behavior change within the affected communities. The results also indicate that the main driver of transmission appeared to be casual sex rather than attendance at recreational clubs frequented by the MSM community. We additionally found that contact tracing was able to approximately halve the total number of cases and that increasing the coverage of

traced contacts might be highly effective for control. Finally, we suggest that new outbreaks could be fully prevented by vaccinating the MSM population at the highest risk.

## Methods

### Epidemiological data

Data for this study were collected within surveillance activities for public health purposes from the Ministry of Health and local health authorities and informed consent was obtained from all participants. Authors affiliated to the Ministry of Health provided the other authors responsible for the analysis with an anonymized line list of confirmed mpox cases, with individual data about age, gender, self-identification as an MSM, dates of possible exposure, date of symptom onset, date of diagnosis, epidemiological investigations, and the existence of other known cases among their contacts at the time of diagnosis. As of December 30, 2022, 921 confirmed cases were recorded; 907 (98.4%) were male and 523 identified as MSM out of the 547 men who disclosed this information (95.6%). The median age was 37 years and half of the cases were between 31 and 44 years old, with range 14–71. The first symptom onset was recorded on May 9, 2022; only one case was asymptomatic and only 3 symptomatic cases did not have a date of symptom onset.

The diagnostic delay (distribution of delay between symptom onset and diagnosis) was found to decrease significantly from a mean of 8 days for cases with symptom onset in the early phase (May 9–June 7), to 6.5 days for cases with later onset (see Supplementary Materials). Of 399 cases for which the information was available, 74 (18.5%, 95%CI of the binomial distribution 14.9–22.7%) were contacts of other known cases. We considered four periods of 30 days each (Period 1: May 9–June 7; Period 2: June 8–July 7; Period 3: July 8 – August 6; Period 4: August 7–September 5) and one period (Period 5) from September 6 to the end of data. Only 40 cases (4.3%) had symptom onset during Period 5.

Information from epidemiological investigations were used to assign cases to one of the following mutually exclusive categories: (i) "household", if the case had another confirmed case in the household at the time of diagnosis; (ii) "club", if the case attended a sauna, sex club, sex party or LGBT pride event in the three weeks preceding symptom onset; (iii) "sex, low activity" and (iv) "sex, high activity", if none of the previous conditions applied and the case reported respectively ≤2 or ≥3 sexual partners in the three weeks preceding symptom onset. This tentative attribution of the location of exposure was used for calibration of the model and was possible for 53% of cases with symptom onset in Period 1, 37% of cases in Period 2 and less than 15% of cases in the following periods.

Vaccination data on the Italian national campaign were provided by the Ministry of Health. The campaign started on August 8, 2022, prioritizing groups of MSM at higher risk of infection and immunized about 13,000 individuals by September 30 (corresponding to about 31% of the estimated HR population).

### Epidemiological model

We simulated the MSM Italian population and their household members using an individual-based model. We assumed that 10% of men between 15 and 70 years old are MSM, based on recent statistics from the United Kingdom[23]. After the exclusion of an age-specific proportion of individuals who did not have a sexual debut with men, according to data reported for Italy in the 2017 European MSM Internet Survey (EMIS)[18], the resulting population of sexually active MSM was 1.67 million. We additionally considered their household members, distributed according to Italian census data[24], for a total modeled population of 5.3 million. We simulated the network of sexual partnerships among MSM based on the numbers of steady and non-steady yearly sexual partners reported for Italy in the EMIS-2017 data[18]. MSM was first assigned a number of potential sexual partners throughout one year (degree of each node in the network). Potential partnerships (edges of the network) were then assigned at random, in the baseline analysis. In two sensitivity analyses, we considered assortative sexual networks where MSM were more likely to be partners of individuals with (a) similar numbers of sexual partners, or (b) both similar numbers of sexual partners and similar age. In absence of adequate data, the model does not distinguish between repeated one-off interactions and recurring sexual partnerships (see Supplementary Material). For nomenclature convenience, the MSM population was divided in 5 risk groups based on their number of yearly sexual partners (Table 1). In the baseline analysis, HR-MSM were assigned exactly 30 potential partners in the sexual network but were allowed to have additional infectious contacts by attending MSM recreational clubs. The choice for allowing only HR-MSM in recreational clubs was based on the limited total capacity of such venues in Italy (see Supplementary Material). To evaluate the effects of deviations from these assumptions, we run two further sensitivity analyses: (c) we did not set a limit to the effective number of yearly partners in the sexual contact network while maintaining the restriction of club attendance to HR-MSM only; (d) we allowed any MSM to attend clubs until occupancy reached club capacity.

Transmission in the model can occur over three routes: (i) within households (irrespectively of whether it happens via sexual intercourse between cohabiting partners or via fomites); (ii) over the sexual network; (iii) within clubs. Transmission dynamics were modeled through an SEIRV (Susceptible-Exposed-Infectious-Removed-Vaccinated) structure, where exposed individuals (E) are non-infectious and become infectious (I) after an incubation period that was sampled from a distribution previously estimated from Italian data (mean 9 days)[10]. Asymptomatic cases[22,25] and possible pre-symptomatic transmission[21] are not explicitly modeled. Undiagnosed infectious individuals remain so for an exponentially distributed duration of average 20 days, based on viral shedding data[26]. Individuals may self-report symptoms based on a probability estimated during calibration. In case of self-reporting, diagnosis occurs after a delay since the date of symptom onset, whose distribution was estimated from observed data before and after June 8. For each diagnosed case, we assume full and effective isolation until healing, and that all household members and 20% of their sexual contacts, not occurring within clubs, are successfully traced. Diagnosis of infectious traced contacts occurs between 1 and 3 days after the latest between the date of diagnosis of the index case and the date of symptom onset of the secondary case. At the time of diagnosis, the classification used for epidemiological data is simulated for all cases in the model using the same criteria. Susceptible individuals who were vaccinated were assumed to have a reduction in the probability of being infected of 80% after 14 days from the first dose and of 90% after 42 days (to simulate the ramp-up of a second dose administered 28 days after the first)[12]. No therapeutic effect was assumed when the vaccine was administered to exposed or infectious individuals.

### Calibration

To calibrate the six free model parameters (transmission rates in household, sexual network and clubs; ratio of self-reporting per infection; and the reductions of transmission rates in sexual network and clubs after June 8), we applied Approximate Bayesian Computing based on Sequential Monte Carlo (ABC-SMC)[27]. We used as a score function the log of the likelihood composed by the product of two components: (a) the Poisson likelihood of the observed total number of cases over the first four epidemiological periods; and (b) the product of the multinomial likelihood of the observed cases by assigned category in the first two epidemiological periods. We performed $T = 2$ sequential steps, with 100,000 parameter sets sampled uniformly and independently from a broad range of values in the first step, and applying a gaussian perturbation kernel on selected parameters in the second step. In the first and second step respectively, we set as

acceptance thresholds 3 times and 2 times the marginal log-likelihood (see Supplementary Material). A total of 694 simulations were selected after the second step, corresponding to 488 unique parameter sets representing an approximation of the joint posterior distribution of parameters. Results for counterfactual scenarios are presented by pooling together 50 stochastic replicates for each of 100 samples without replacement from the joint posterior distribution of parameters (total 5000 simulations for each scenario).

### Reporting summary

Further information on research design is available in the Nature Portfolio Reporting Summary linked to this article.

## Data availability

The mpox surveillance data generated in this study have been deposited in the Zenodo database under accession code 10697578.

## Code availability

The code for the mpox Individual Based Model used in this manuscript has been developed in C using GNU Scientific Libraries version 2.7.1. The code was deposited in the Zenodo database under accession code 10697578.

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

## Acknowledgements

This research was supported by funding from EU grant 101045989 VERDI and EU funding within the NextGenerationEU-MUR PNRR Extended Partnership Initiative on Emerging Infectious Diseases (project no. PE00000007, INF-ACT), both received by SM. AV acknowledges support from CDC contract-75D30122C14810. The findings and conclusions in this study are those of the authors and do not necessarily represent the official position of the funding agencies.

## Author contributions

G.G., G.R., A.V., I.L., M.A., and S.M., designed the study. GG developed the code; G.G. and V.M. and performed the analyses. A.M., A.S., F.F., A.C., F.M. collected, preprocessed and shared the data. G.G., V.M., A.V., I.L., M.A., and S.M. drafted the manuscript. All authors read, reviewed and approved the manuscript.

## Competing interests

The authors declare no competing interests.
