## [Peer Review File · Nature Communications]

The decline of the 2022 Italian mpox epidemic: role of behavior changes and control strategiesREVIEWER COMMENTS

Reviewer #1 (Remarks to the Author):

The paper presents a study of mpox outbreak in Italy using an individual based model capturing MSM sexual contact networks. They present the role of behavioral change in the MSM community as well as contact tracing in the rapid decline of mpox cases. They also highlight potential strategies for preventing mpox resurgences in the future including targeted immunization of high risk groups.

I believe the paper makes an important contribution through a well-calibrated individual-based model and a robust analysis of potential interventions. Below are some questions on certain methodological choices and details.

Comments:

Although perhaps unintended, the paper pits the targeted immunization of high risk groups with ring vaccination strategies in the presence of contact tracing. It is not clear if the former assumes baseline contact tracing for new outbreaks. Further, more details are needed on the activity modeling and how recurring sexual contacts are distinguished from one-off interactions (not just in club settings).

The simulation period spans about 9 months. Were the edges to yearly sexual partners equally distributed or always present in the network? Also club attendance (which includes LGBT pride events noted as a potential source of early cases) is restricted to those with 30+ sexual partners. Not enough justification is provided for this choice.

Page 4: line 121: The authors mention both reduction in number of sexual contacts and the reduction of transmissibility (figures show the latter). Is this meant to be the same parameter summarized differently? One could imagine change in per edge transmissibility through safe sex practices without changing the number of edges. Would be good to differentiate these and clarify what is being modified in the model. Likewise, how was club attendance modulated? From Figure 3, the median R_t in club settings was higher than in the sexual contact network (although with a large uncertainty). Is this a result of fewer individuals with high contact?

It would be good to have one-line definitions of the mitigation strategies CT, RT, CV, RV. For example, mention that the ring strategies are focused on contacts of contacts.

From Table 1, MR and HR population totally accounts for 23% of the sexually active MSM. In other places this is mentioned as 26%. Also, were the results in Figure 6 obtained in the baseline setting (with contact tracing at 20%) or with no other active interventions?

Consider rephrasing line 216 (page 6). the word "even" in the phrase "even in the absence of any vaccination" seems out of place.

Reviewer #2 (Remarks to the Author):

Review of "Evaluating the role of behavior changes and control strategies in the decline of the mpox 2022 epidemic in Italy: a modeling study"

This is a modelling study on the transmission of Mpox in Italy during the significant outbreak in 2022. The goal of the authors was to quantify how much behaviour change among MSM people in Italy contributed to the turning over of the epidemic in July 2022 whilst accounting for the effect of the roll-out of prophylactic vaccination (e.g. Imvanex/Jynneos) to at-risk communities, and the effect of susceptible depletion. The authors used an individual based transmission model (IBM) to

simulate the Mpox infection process within households containing a mix of MSM and non-MSM individuals, as well as among the sexual contacts of MSM individuals including sex-on-premises clubs. The demographics and geography of the modelled population were carefully calibrated to Italian population data.

Overall, I think this is a valuable modelling study. Constructing an IBM to simulate Mpox spread has allowed the authors to directly model the effect of social (network) structure on transmission dynamics as well as the effect of contact tracing. This gives a distinct perspective compared to other modelling approaches in the literature which include mathematical approximations to network dynamics that exclude intervention or behaviour change (Murayama et al 2023; *The Journal of Infectious Diseases*), and heterogeneous group mass-action models (e.g. Brand et al 2023; *Nature Comms.*, Xiridou et al 2023; *MedRxiv*) which don't implement contact tracing directly.

However, I do have some issues which range across the statistical inference, epidemiological model choices and presentation of results.

Issue 1:

I don't understand why the proportion of sexual contacts that are contact traced is fixed at 20%? Surely, this is a key unknown and should be included among the parameters for Bayesian inference?

Issue 2:

In Figure 1 the agreement between the mean over model simulations using parameters from the (approximate) posterior distribution and the data is reasonable. However, the simulated epidemic curves appear highly over-dispersed, for example, all data points are deep within the 95% prediction intervals. This seems to indicate mis-calibration of the parameters, which could be diagnosed using a probability integral transform histogram.

For Bayesian inference with stochastic epidemic models it is useful to view Bayesian inference as sampling from the joint conditional distribution of both the parameters, the unobserved infections and other unobserved quantities such as the underlying sexual contact graph:

$\text{Prob}(\text{parameters, unobserved_infections, unobserved_contact_structure} \mid \text{observed_data})$

When diagnosing model calibration (e.g. Figure 1) it is better to use this posterior distribution since this is the distribution being inferred by ABC (operationally this can be done by capturing the sampled epidemic trajectories jointly with parameter sets when accepting).

On the other hand, when considering counter-factual scenarios from the beginning of the epidemic (e.g. Figure 4) it is more intuitive to simulate using parameters drawn from the marginal posterior distribution $P(\text{parameters} \mid \text{observed_data})$ as the authors of this paper have done. This has the interpretation of comparing the distribution of outcomes over many different possible ways the outbreak could have developed informed by the single trajectory observed.

It might be a good idea to show figure 1 using the inferred unobserved infections.

Issue 3:

It appears that only 125 parameter sets are sampled from Rejection-ABC, which means that there is low confidence in the reported 95% credible intervals of the parameters. The authors produce more epidemic trajectories (50 per sampled parameter set), but this doesn't fix this problem. I realise that this is a computationally intensive model to sample from. A possible solution is to construct an acceptance threshold as described in the paper, but then use a more efficient ABC technique such as ABC-SMC (e.g. Toni et al 2009; *Journal of the Royal Society Interface*).

Issue 4:

I understand that this model does not include pre/a – symptomatic infections. The problem with this is that then the mean generation period in this model is stuck somewhere between 9 – 29 days depending on the proportion of self-reporting cases and how quickly they cease making infectious contacts after symptom development. However, for contact traced cases in the UK the mean serial interval was 8 days (Ward et al 2022; BMJ also cited by the authors). Having potentially a much longer mean period between infector and infectee than actual can bias results by causing mis-estimation in reproductive numbers.

For example, a model with an (incorrect) long mean generational interval would require a larger drop in the reproductive number to achieve an observed decline curve in an epidemic compared to a model with a (correct) shorter mean generation interval.

I would like the authors to consider how to mitigate the effect of (possibly overly) long mean generational intervals on the parameter inference.

Issue 5:

The code for the model provided does run and produce outputs, however, I would expect scripts for plotting the output (i.e. reproducing the paper plots) would be provided which I could then run on the output file structure. Moreover, I don't see that a script for running the ABC has been provided.

The documentation of the C functions is patchy, some have doc strings in the body of the function, some don't.

Could the authors include a script to reproduce the paper plots from the model runs (did you use a fixed random seed?) and the script used to run the ABC?

Minor Issue 1:

I find the choice of sexual contact network construction slightly unusual. The authors consider both a random and assortative selection over individuals, however, neither are common choices in the literature of sexual contact graphs that I'm aware of. The most common model seems to be the random partnership model, where each individual has a number of half-edges according to their generated number of sexual partners, and the half-edges are paired at random. This naturally favours connection to high degree individuals according to a size-biased distribution.

Could the authors either further justify their choice of graph generation algorithm or explore using the random partnership model?

Minor Issue 2:

I'm not sure what R_t is in figure 3 or in the main text, for heterogeneous populations the definition of the Reproductive number can be a bit tricky, could the authors make their definition clearer?

Minor Issue 3:

The results of this paper differ from the other modelling papers I've seen on Mpox which either place no role on behavioural change (Murayama et al 2023; The Journal of Infectious Diseases), or find a less significant role comparable to population immunity (Brand et al 2023; Nature Comms., Xiridou et al 2023; MedRxiv).

In discussion, could the authors comment on why this might be? Is this an effect of the more realistic contact structure in this model?

Reviewer #3 (Remarks to the Author):

This study uses an agent-based modeling approach to simulate features of the 2022 mpox outbreak. The authors suggest their results indicate that a substantial reduction in sexual activity occurred in June of 2022 that helps explain the course of the outbreak. They also find that vaccinating contacts or tracing secondary contacts would have only a small impact on the outbreak.

These results somewhat differ from other simulation studies. It would be extremely helpful if the authors further engaged with other modeling work such as Zhang et al. (2023) in *Lancet Infectious Disease* and Spicknall et al., 2022 in *MMWR*. In particular these two studies seem to indicate a reduction in sexual activity (~40 - 50%) that is substantially Lower than the current observations.

The choice of modeling "behavior change" through the a free parameter of "reduction in transmissibility" over a static sexual network is somewhat confusing to me. For example, the authors state, "A massive reduction in the number of sexual contacts... and in the general sexual network... is necessary for the model to reproduce the observed trends in data." Yet, the model, and results in Figure 2, appear to be merely estimating the change the transmissibility per sexual contact not explicitly estimating a changing number of sexual contacts, which appears to be static through the simulation - according to my understanding of the supplemental material. That is, is a percentage reduction in the per-partner change in transmissibility the same thing as a percentage reduction in sexual partners? Does it mean the same thing for disease control? This could still be a misreading but, even if so, it should be clarified and made clear to readers.

In addition, despite utilizing data on sexual partnerships to estimate levels of sexual activity, the specifics regarding some modeling decisions surrounding sexual partnerships and club attendance are not very clear. Additional detail, even if only included in the supplemental material would be extremely beneficial. For example, why were sexual partnerships capped at exactly 30? Why were individuals with these number of partners the only ones assigned to attend bars? Are there data to back up these assumptions or were they made for practical reasons.

Finally, while it is beneficial that the authors conducted a sensitivity analysis examining the impact of assortative mixing by sexual activity, decades of research has provided substantial evidence of the non-random mixing of sexual partnerships - particularly in MSM - by categories such as age, ethnicity, etc. These factors could also interact with other important features of this simulation such as level of sexual activity, club attendance, etc. Randomly assigned sexual partners seems an extreme oversimplification of this process. The authors should provide additional justification for this decision, provide evidence this is not the case amongst MSM in Italy, or show evidence that it does not impact their conclusions.

REVIEWER COMMENTS

Reviewer #1 (Remarks to the Author):

The paper presents a study of mpox outbreak in Italy using an individual based model capturing MSM sexual contact networks. They present the role of behavioral change in the MSM community as well as contact tracing in the rapid decline of mpox cases. They also highlight potential strategies for preventing mpox resurgences in the future including targeted immunization of high risk groups.

I believe the paper makes an important contribution through a well-calibrated individual-based model and a robust analysis of potential interventions. Below are some questions on certain methodological choices and details.

We thank the reviewer for their appreciation of our work and the constructive feedback.

Comments:

Although perhaps unintended, the paper pits the targeted immunization of high risk groups with ring vaccination strategies in the presence of contact tracing. It is not clear if the former assumes baseline contact tracing for new outbreaks.

We apologize for the lack of clarity. All immunization strategies assume baseline contact tracing, since we assume that, in the case of a new outbreak, this intervention would be maintained even after a successful immunization campaign. We have now made this explicit in the text (lines 213-215). As the reviewer suggests, the evaluation of ring vaccination strategies, improved contact tracing, and immunization strategies was meant to assess independently possible ways of improving control of such outbreaks and not to pit one strategy against the other. On the opposite, a combination of strategies would be more beneficial to minimize the risk of onward transmission for emerging pathogens. We now rephrased the Results and Discussion in such a way to avoid suggesting a competition among strategies.

Further, more details are needed on the activity modeling and how recurring sexual contacts are distinguished from one-off interactions (not just in club settings).

We apologize for the lack of details. In absence of available data on for Italy, we did not explicitly model one-off interactions vs. recurring sexual contacts. More in general, individual sex acts are not explicitly represented, given that available data for Italy do not contain information on the frequency/number of sex acts, but only on the number of yearly sexual partners. We now made this explicit in the Methods (lines 404-406) and in the Supplementary Material (Section 1.5).

The simulation period spans about 9 months. Were the edges to yearly sexual partners equally distributed or always present in the network?

We apologize for the lack of clarity. The network was built at the start of each simulation based on the number of yearly sexual contacts, without scaling for the duration of the simulation, i.e., edges were always present in the network. At each time step of the simulation, an infectious individual has a probability of infecting each of its assigned sexual partners depending on a transmissibility parameter β_S , which is calibrated against epidemiological data. We now described this more explicitly in the Methods section (lines 399-402) and in the Supplementary Material (Section 1.2).

Also club attendance (which includes LGBT pride events noted as a potential source of early cases) is restricted to those with 30+ sexual partners. Not enough justification is provided for this choice.

Our choice to restrict club attendance to individuals with 30 or more yearly sexual partners was based on the limited number and total capacity of such clubs in Italy (50 clubs with an overall capacity of approximately 7,500, corresponding to ~0.4% of the total modelled MSM population), which implies a very small size of the potential customer population. Given the nature and purpose of such clubs, it is reasonable to assume that customers of clubs are strongly over-represented among the highest activity groups. When considering that only individuals with 30+ yearly sexual partners (2.5% of the population) are potential club customers, existing clubs will be fully attended if $0.4\%/2.5\% = 16\%$ of all potential customers will attend a club in a given week, resulting in an average frequency of attendance to clubs equal to once every 6 weeks. Extending the potential customer population to lower sexual activity groups would imply a much lower average frequency of attendance to clubs: for example, if we include all individuals with 10+ sexual partners, the average attendance frequency would be once every year. We now report a more extensive explanation of this assumption in the Methods (lines 410-411) and Supplementary Material (Section 1.3).

Moreover, to evaluate the robustness of our results with respect to this assumption, we added a sensitivity analysis where we allowed any MSM to attend a club in a given week, irrespectively of their number of yearly sexual partners, until club capacity is reached. After recalibration, transmission of mpox in clubs was estimated to be much lower compared to the main analysis, with a mean reproduction number in clubs of 1.3 (against 2.7 in the main analysis) and a mean proportion of cases acquired in clubs of 2.5% (against 10% in the main analysis). The lower importance of club transmission explains why the model estimates a smaller reduction of transmissibility in clubs (mean 36% vs. 87% in the main analysis) and requires a higher reduction of transmissibility in the sexual contact network (mean 78% vs. 61% in the main analysis). A higher attack rate was estimated in lower sexual activity groups and a lower attack rate in the higher sexual activity groups, supporting the conclusion that population immunity in high-risk groups did not contribute significantly to the downturn of the epidemics. We now report the methods and results of the sensitivity analysis in the Supplementary Material (Section 4.4).

Page 4: line 121: The authors mention both reduction in number of sexual contacts and the reduction of transmissibility (figures show the latter). Is this meant to be the same parameter summarized differently? One could imagine change in per edge transmissibility through safe sex practices without changing the number of edges. Would be good to differentiate these and clarify what is being modified in the model. Likewise, how was club attendance modulated?

We apologize for the confusion and the poor wording. In our model, the force of infection in the sexual contact network and in clubs depends on a transmissibility parameter (β_S and β_C respectively) which accounts at the same time for the probability of transmission per sex act and for the frequency of sex acts with potential sexual partners. The reduction in the force of infection is encoded by two scaling parameters (χ_S and χ_C respectively) for the transmissibility parameters. The edges in the sexual network (number of yearly sexual partners) and the number of potential club customers do not change throughout the simulation horizon. Each parameter χ can be interpreted either as a reduction in the frequency of sex acts (in the network or in clubs), or as a reduction in the probability of transmission per sex act, or as a combination of both effects. Although we cannot quantify the relative contributions of the two components in the reduction of the force of infection, our results suggest that behavior change seems to be the main driver of the decline of mpox in Italy. We now rephrased all related text (lines 28, 126-128, 140-141, 149-150, 155-156, 243-253 and Section 1.5 of the Supplementary Material), in such a way to avoid confusion or misinterpretation of our results. We thank the reviewer for the useful comment.

From Figure 3, the median Rt in club settings was higher than in the sexual contact network (although with a large uncertainty). Is this a result of fewer individuals with high contact?

The Rt in setting X for period P was computed as the ratio $\frac{c_{2,X}}{c_{1,X}}$, where $c_{1,X}$ is the total number of primary cases infected within period P who participate in setting X (i.e., all infected individuals except those who do not attend clubs for club transmission and those who live alone for household transmission) and $c_{2,X}$ is the total number of secondary cases caused in setting X by all individuals counted in $c_{1,X}$. The higher Rt in club settings depends on the estimated transmissibility parameter β_C and, as the reviewer noticed, can be interpreted as due to a high number of sexual contacts within clubs. The reason why this high reproduction number does not convert into a high proportion of mpox cases due to club transmission in the general population is the small number of individuals attending clubs. The large variability derives from both the variability of the posterior distribution of β_C and from the small number of infectors in clubs $c_{1,C}$ during period 1. We now added the description of the computation of the reproduction number in the Supplementary Material (Section 3.1).

It would be good to have one-line definitions of the mitigation strategies CT, RT, CV, RV. For example, mention that the ring strategies are focused on contacts of contacts.

We have now added one-line definitions of these strategies in the figure caption. We thank the reviewer for this useful suggestion.

From Table 1, MR and HR population totally accounts for 23% of the sexually active MSM. In other places this is mentioned as 26%.

We apologize for the oversight. The correct figure is 23% and it was now corrected throughout the text.

Also, were the results in Figure 6 obtained in the baseline setting (with contact tracing at 20%) or with no other active interventions?

We apologize for the lack of clarity. As explained above, Figure 6 was obtained with baseline contact tracing at 20%. This is now made explicit.

Consider rephrasing line 216 (page 6). the word “even” in the phrase “even in the absence of any vaccination” seems out of place.

The whole sentence has been dropped in response to another comment. We thank the reviewer for the careful reading.

Reviewer #2 (Remarks to the Author):

Review of “Evaluating the role of behavior changes and control strategies in the decline of the mpox 2022 epidemic in Italy: a modeling study”

This is a modelling study on the transmission of Mpox in Italy during the significant outbreak in 2022. The goal of the authors was to quantify how much behaviour change among MSM people in Italy contributed to the turning over of the epidemic in July 2022 whilst accounting for the effect of the roll-out of prophylactic vaccination (e.g. Imvanex/Jynneos) to at-risk communities, and the effect of susceptible depletion. The authors used an individual based transmission model (IBM) to simulate the Mpox infection process within households containing a mix of MSM and non-MSM individuals, as well as among the sexual contacts of MSM individuals including sex-on-premises clubs. The demographics and geography of the modelled population were carefully calibrated to Italian population data.

Overall, I think this is a valuable modelling study. Constructing an IBM to simulate Mpox spread has allowed the authors to directly model the effect of social (network) structure on transmission dynamics as well as the effect of contact tracing. This gives a distinct perspective compared to other modelling approaches in the literature which include mathematical approximations to network dynamics that exclude intervention or behaviour change (Murayama et al 2023; The Journal of Infectious Diseases), and heterogeneous group mass-action models (e.g. Brand et al 2023; Nature Comms., Xiridou et al 2023; MedRxiv) which don't implement contact tracing directly.

We thank the reviewer for their appreciation of our work and the constructive feedback.

However, I do have some issues which range across the statistical inference, epidemiological model choices and presentation of results.

Issue 1:

I don't understand why the proportion of sexual contacts that are contact traced is fixed at 20%? Surely, this is a key unknown and should be included among the parameters for Bayesian inference?

We fixed this parameter to reduce the complexity of model calibration. Individual based models are computationally demanding and increasing the dimensionality of the parameter space increases non-linearly the computational time required for calibration. Furthermore, including this parameter would require using additional data, such as the observed number of cases that were contacts of known cases, in the computation of the goodness of fit. How to weight the importance of this heterogeneous data point relative to the data already considered for the goodness of fit (case counts by period and category) is not trivial. For these reasons, we preferred to fix the value at 20% based on early simulations and then to validate this choice against the observed number of cases that were contacts of known

cases (Fig. 1d). To further demonstrate the goodness of our choice, we add a figure below and in the Supplementary Material (Figure S4) where we compare the model-estimated proportion obtained when considering alternative values of the contact tracing coverage. The grey horizontal line and gray shaded area represent the mean and 95%CI of the observed proportion.

Figure S4. Percentage of cases diagnosed via contact tracing under different contact tracing coverage values. Boxplots represents the mean (central bar), interquartile range (IQR, rectangular box) and 95% prediction intervals (PI) (whiskers); the gray horizontal dashed line and shaded area represent the mean and 95% confidence interval (CI) of the binomial distribution for the probability that a diagnosed case is a contact of another case in observed data.

Issue 2:

In Figure 1 the agreement between the mean over model simulations using parameters from the (approximate) posterior distribution and the data is reasonable. However, the simulated epidemic curves appear highly over-dispersed, for example, all data points are deep within the 95% prediction intervals. This seems to indicate mis-calibration of the parameters, which could be diagnosed using a probability integral transform histogram. For Bayesian inference with stochastic epidemic models it is useful to view Bayesian inference as sampling from the joint conditional distribution of both the parameters, the unobserved infections and other unobserved quantities such as the underlying sexual contact graph:

```
Prob(parameters, unobserved_infections, unobserved_contact_structure |  
observed_data)
```

When diagnosing model calibration (e.g. Figure 1) it is better to use this posterior distribution since this is the distribution being inferred by ABC (operationally this can

be done by capturing the sampled epidemic trajectories jointly with parameter sets when accepting).

We thank the reviewer for this useful comment. We now show Fig1c using only the epidemic trajectories accepted jointly with parameter sets and report the corresponding probability integral transform histogram in the Supplementary Material (Figure S7) and below. Please note that only the period-specific totals, not the daily epidemic trajectory, were used during calibration. The residual discrepancies between modeled epidemic curves and data may be attributable to the simplification of behavior change as happening all at once rather than gradually.

Figure S7. Histogram of Probability Integral Transform values for marginal predictions of daily cases. The dashed horizontal line represents the target uniform distribution.

On the other hand, when considering counter-factual scenarios from the beginning of the epidemic (e.g. Figure 4) it is more intuitive to simulate using parameters drawn from the marginal posterior distribution $P(\text{parameters} \mid \text{observed_data})$ as the authors of this paper have done. This has the interpretation of comparing the distribution of outcomes over many different possible ways the outbreak could have developed informed by the single trajectory observed. It might be a good idea to show figure 1 using the inferred unobserved infections.

We added a new panel (e) in Figure 1 where we report the distribution of unobserved infections estimated in each of the accepted simulations (median 231, 90%CI: 9-5983). The variability found in the number of unobserved infections depends on the variability of accepted values for the probability of self-reporting (Figure 2a).

Issue 3:

It appears that only 125 parameter sets are sampled from Rejection-ABC, which means that there is low confidence in the reported 95% credible intervals of the parameters. The authors produce more epidemic trajectories (50 per sampled parameter set), but this doesn't fix this problem. I realise that this is a computationally intensive model to sample from. A possible solution is to construct

an acceptance threshold as described in the paper, but then use a more efficient ABC technique such as ABC-SMC (e.g. Toni et al 2009; Journal of the Royal Society Interface).

Following the reviewer's suggestion, we now use an ABC-SMC calibration technique with two steps. We now selected 694 simulations from 488 unique parameter sets in the baseline analysis. Full details on the implemented ABC-SMC procedure are reported in the Methods (lines 442-458) and Supplementary material (Section 2). We thank the reviewer for this important suggestion which allowed us to improve the robustness of parameter estimation.

Issue 4:

I understand that this model does not include pre/a – symptomatic infections. The problem with this is that then the mean generation period in this model is stuck somewhere between 9 – 29 days depending on the proportion of self-reporting cases and how quickly they cease making infectious contacts after symptom development. However, for contact traced cases in the UK the mean serial interval was 8 days (Ward et al 2022; BMJ also cited by the authors). Having potentially a much longer mean period between infector and infectee than actual can bias results by causing mis-estimation in reproductive numbers. For example, a model with an (incorrect) long mean generational interval would require a larger drop in the reproductive number to achieve an observed decline curve in an epidemic compared to a model with a (correct) shorter mean generation interval.

I would like the authors to consider how to mitigate the effect of (possibly overly) long mean generational intervals on the parameter inference.

The serial interval and generation time distributions are usually estimated through contact-tracing data and may depend significantly on both the structure of the transmission network and the intensity of the intervention. Earlier estimates for the mean generation time in Italy were about 12.5 days, with 2.5 and 97.5 percentiles at 5 and 23 days [Guzzetta et al., Em Inf Dis. 2022]. To assess whether the generation time was correctly reproduced, we computed the generation time for all detected secondary cases in the simulations as the difference between the known dates of infection of the case itself and of its infector. We obtained a mean generation time of 14.4 days (95%CI across selected simulations within ABC-SMC: 14.3-14.6) and a 2.5 and 97.5 percentile of 5 and 32 days respectively. The modeled distribution and the one derived from data are graphically compared in the figure reported below and in the Supplementary Material (Figure S8). The slightly longer mean and 97.5 percentile estimated by the model may not necessarily be an over-estimation, given that contact tracing data in real life may miss some of the longer generation times due to recall biases or difficulty in tracing contacts who are more distant in time. Given these results, we believe that overly long generation intervals are not a significant issue in our work. We have now added a sentence reporting this further validation of the model in the Discussion (lines 305-310) with all methodological details in the Supplementary Material (Section 3.2).

Figure S8. Modeled distribution of the realized generation times across simulations accepted during calibration and distribution obtained from the mean parameters estimated from Italian contact tracing data [Guzzetta et al., 2022].

Issue 5:

The code for the model provided does run and produce outputs, however, I would expect scripts for plotting the output (i.e. reproducing the paper plots) would be provided which I could then run on the output file structure. Moreover, I don't see that a script for running the ABC has been provided.

The documentation of the C functions is patchy, some have doc strings in the body of the function, some don't.

Could the authors include a script to reproduce the paper plots from the model runs (did you use a fixed random seed?) and the script used to run the ABC?

We apologize for the poor reproducibility of our results. We are now uploading a zip file containing the new version of the C code, bash scripts for running ABC-SMC and scenario simulations, R scripts for reading results and plotting figures, and an updated README file with instructions for running the analyses.

Minor Issue 1:

I find the choice of sexual contact network construction slightly unusual. The authors consider both a random and assortative selection over individuals, however, neither are common choices in the literature of sexual contact graphs that I'm aware of. The most common model seems to be the random partnership model, where each individual has a number of half-edges according to their generated number of sexual partners, and the half-edges are paired at random. This naturally favours connection to high degree individuals according to a size-biased distribution.

Could the authors either further justify their choice of graph generation algorithm or explore using the random partnership model?

We apologize for the lack of clarity. The network generation algorithm in the main analysis is essentially a random partnership model: the number of yearly sexual partners (half-edges) are first assigned to each individual according to the distribution estimated from empirical data, and the potential partnerships (pairing of half-edges) are assigned at random, naturally favoring connection among high-degrees individuals. However, the literature of sexually transmitted infections suggests that contacts may be even more assortative by sexual activity level than in a random partnership model (i.e. that individuals tend to have a stronger preference to encountering other individuals with a similar level of sexual activity). For this reason, we introduced a sensitivity analysis where we attempt reproducing this effect. Please note that, based on a suggestion from another reviewer, in the revised manuscript we also include a further sensitivity analysis where assortativity by both sexual activity and age is considered. The new sensitivity analysis supports our conclusions. We now specify more explicitly the method for the generation of the sexual contact network in the Methods (lines 399-406) and in the Supplementary Material (Sections 1.2 and 4.1-4.2 for the Sensitivity analyses).

Minor Issue 2:

I'm not sure what R_t is in figure 3 or in the main text, for heterogeneous populations the definition of the Reproductive number can be a bit tricky, could the authors make their definition clearer?

We apologize for the lack of clarity. Estimates of the reproduction number in the model were computed as the ratio $\frac{c_{2,X}}{c_{1,X}}$, where $c_{1,X}$ is the total number of primary cases infected within period P who participate in setting X (i.e., all infected individuals except those who do not attend clubs for club transmission and those who live alone for household transmission) and $c_{2,X}$ is the total number of secondary cases caused in setting X by all individuals counted in $c_{1,X}$. We now added the description of the computation of the reproduction number in the Supplementary Material (Section 3.1).

Minor Issue 3:

The results of this paper differ from the other modelling papers I've seen on Mpox which either place no role on behavioural change (Murayama et al 2023; The Journal of Infectious Diseases), or find a less significant role comparable to population immunity (Brand et al 2023; Nature Comms., Xiridou et al 2023; MedRxiv).

In discussion, could the authors comment on why this might be? Is this an effect of the more realistic contact structure in this model?

We thank the reviewer for this useful comment. We now added in the Discussion a paragraph comparing our conclusions with those from other models (lines 689-702).

The differences between our study and other published ones may arise from country-specific transmission dynamics due to the underlying sexual contact structure and intensity and effectiveness of contact tracing, as well as from model assumptions. Two studies [Brand et al. Nat Com 2023, Zhang et al., Lan Inf Dis 2023] investigated the UK outbreak, where a four-fold cumulative incidence of diagnosed cases was observed compared to Italy and where the sexual contact network seems characterized by a stronger representation of high-activity groups. Both factors contribute to a higher cumulative incidence (and therefore higher population immunity) in high-activity groups; in these conditions, a smaller reduction of behavioral change would be sufficient to explain the downturn of the curve. In addition, the shorter serial interval identified in UK contact tracing data compared to Italy [Ward et al., 2022; Guzzetta et al., 2022] may suggest a higher ability in tracing cases rapidly in the UK, which may have also contributed to support the end of the epidemic with smaller amounts of behavior changes. Similarly, the mpox epidemics in the Netherlands [Xiridou et al., medRxiv 2023] was analyzed in a preprint where the population is divided in four sexual activity levels, of which the two highest risk groups have more than 50 casual partners per year and represent 18% of the MSM population (compared to 2.5% with 30+ partners in Italian data); in addition, they assume sexual activity groups to mix in a strongly assortative way, with 70-80% of all sexual contacts occurring among people from the same group. This may justify the decline of mpox as almost exclusively due to the accrual of immunity in high activity groups. The papers by Murayama et al. [J Inf Dis 2023] and Endo et al. [Science, 2022] modeled the epidemic in several countries using the same UK sexual contact data for all countries and without accounting for either behavior change or the effect contact tracing. For Italy, they needed to assume a very small MSM population, amounting to approximately 1.2% of the total male population (including the non-sexually active), compared to, e.g., 1.9% assumed for UK and 2.4% for the Netherlands. Finally, the study by Spicknall et al. [Morb Mort Weekly Rep, 2022] evaluated the theoretical effect of an assumed reduction in sexual activity on a generic population, without specific reference to an observed epidemics and therefore their conclusions cannot be compared to our study.

Reviewer #3 (Remarks to the Author):

This study uses an agent-based modeling approach to simulate features of the 2022 mpox outbreak. The authors suggest their results indicate that a substantial reduction in sexual activity occurred in June of 2022 that helps explain the course of the outbreak. They also find that vaccinating contacts or tracing secondary contacts would have only a small impact on the outbreak.

These results somewhat differ from other simulation studies. It would be extremely helpful if the authors further engaged with other modeling work such as Zhang et al. (2023) in *Lancet Infectious Disease* and Spicknall et al., 2022 in *MMWR*. In particular these two studies seem to indicate a reduction in sexual activity (~40 - 50%) that is substantially Lower than the current observations.

We thank the reviewer for this useful comment. We now added in the Discussion a paragraph comparing our conclusions with those from other models (lines 689-702), including those mentioned by the reviewer. The differences between our study and other published ones may arise from country-specific transmission dynamics due to the underlying sexual contact structure and intensity and effectiveness of contact tracing, as well as from model assumptions. Two studies [Brand et al. *Nat Com* 2023, Zhang et al., *Lan Inf Dis* 2023] investigated the UK outbreak, where a four-fold cumulative incidence of diagnosed cases was observed compared to Italy and where the sexual contact network seems characterized by a stronger representation of high-activity groups. Both factors contribute to a higher cumulative incidence (and therefore higher population immunity) in high-activity groups; in these conditions, a smaller reduction of behavioral change would be sufficient to explain the downturn of the curve. In addition, the shorter serial interval identified in UK contact tracing data compared to Italy [Ward et al., 2022; Guzzetta et al., 2022] may suggest a higher ability in identifying cases rapidly in the UK, which may have also contributed to support the end of the epidemic with smaller amounts of behavior changes. Similarly, the mpox epidemics in the Netherlands [Xiridou et al., *medRxiv* 2023] was analyzed in a preprint where the population is divided in four sexual activity levels, of which the two highest risk groups have more than 50 casual partners per year and represent 18% of the MSM population (compared to 2.5% with 30+ partners in Italian data); in addition, they assume sexual activity groups to mix in a strongly assortative way, with 70-80% of all sexual contacts occurring among people from the same group. This may justify the decline of mpox as almost exclusively due to the accrual of immunity in high activity groups. The papers by Murayama et al. [*J Inf Dis* 2023] and Endo et al. [*Science*, 2022] modeled the epidemic in several countries using the same UK sexual contact data for all countries and without accounting for either behavior change or the effect contact tracing. For Italy, they needed to assume a very small MSM population, amounting to approximately 1.2% of the total male population (including the non-sexually active), compared to, e.g., 1.9% assumed for UK and 2.4% for the Netherlands. Finally, the study by Spicknall et al. [*Morb Mort Weekly Rep*, 2022] evaluated the theoretical effect of an assumed reduction in sexual activity on a generic population, without specific reference to an

observed epidemics and therefore their conclusions cannot be compared to our study.

The choice of modeling “behavior change” through the a free parameter of “reduction in transmissibility” over a static sexual network is somewhat confusing to me. For example, the authors state, “A massive reduction in the number of sexual contacts... and in the general sexual network... is necessary for the model to reproduce the observed trends in data.” Yet, the model, and results in Figure 2, appear to be merely estimating the change the transmissibility per sexual contact not explicitly estimating a changing number of sexual contacts, which appears to be static through the simulation - according to my understanding of the supplemental material. That is, is a percentage reduction in the per-partner change in transmissibility the same thing as a percentage reduction in sexual partners? Does it mean the same thing for disease control? This could still be a misreading but, even if so, it should be clarified and made clear to readers.

We apologize for the confusion and the poor wording. In our model, the force of infection in the sexual contact network and in clubs depends on a transmissibility parameter (β_S and β_C respectively) which accounts at the same time for the probability of transmission per sex act and for the frequency of sex acts with potential sexual partners. The reduction in the force of infection is encoded by two scaling parameters (χ_S and χ_C respectively) for the transmissibility parameters. The edges in the sexual network (number of yearly sexual partners) and the number of potential club customers do not change throughout the simulation horizon. Each parameter χ can be interpreted either as a reduction in the frequency of sex acts (in the network or in clubs), or as a reduction in the probability of transmission per sex act, or as a combination of both effects. Although we cannot quantify the relative contributions of the two components in the reduction of the force of infection, our results suggest that behavior change seems to be the main driver of the decline of mpox in Italy. We now rephrased all related text (lines 28, 126-128, 140-141, 149-150, 155-156, 243-253 and Section 1.5 of the Supplementary Material) in such a way to avoid confusion or misinterpretation of our results. We thank the reviewer for the useful comment.

In addition, despite utilizing data on sexual partnerships to estimate levels of sexual activity, the specifics regarding some modeling decisions surrounding sexual partnerships and club attendance are not very clear. Additional detail, even if only included in the supplemental material would be extremely beneficial. For example, why were sexual partnerships capped at exactly 30? Why were individuals with these number of partners the only ones assigned to attend bars? Are there data to back up these assumptions or were they made for practical reasons.

We apologize for the lack of clarity. The number of yearly partners in the sexual contact network was capped at 30 based on the observation that the large majority of the MSM population (97.5%) had less than 30 yearly sexual partners. We note that the actual number of yearly sexual partners in the model may be much higher when considering sexual contacts within clubs: the figure below shows the distribution of individuals by number of yearly sexual partners in the main analysis, accounting for potential sexual partners

in clubs for individuals attending them (shown in red). For the latter, we added the 30 yearly partners assigned when building the sexual contact network to the total population potentially attending clubs assigned to each individual. In practice, the limit in the number of potential sexual partners in the model is very high (up to 31k).

In order to evaluate the robustness of conclusions with respect to the maximum number of yearly sexual partners in the network, we performed an additional sensitivity analysis where we did not assign a maximum number, and we still allowed only individuals with 30 or more yearly sexual partners to attend clubs. After recalibration of model parameters with the same criteria as in the main analysis, results and conclusions remained qualitatively and quantitatively similar (Section 4.3 of the Supplementary Material).

Our choice to restrict club attendance to individuals with 30 or more yearly sexual partners was based on the limited number and total capacity of such clubs in Italy (50 clubs with approximately overall capacity of 7,500, corresponding to ~0.4% of the total modelled MSM population), which implies a very small size of the potential customer population. Please note that the venues that are modeled here are not generic bars catering to the LGBT community but clubs restricted to MSM allowing for sex-on-premises. Given the nature and purpose of such clubs, it is reasonable to assume that customers of clubs are strongly over-represented among the highest activity groups. When considering that only individuals with 30+ yearly sexual partners (2.5% of the population) are potential club customers, existing clubs will be fully attended if $0.4\%/2.5\% = 16\%$ of all potential customers will attend a club in a given week, resulting in an average frequency of attendance to clubs equal to once every 6 weeks. Extending the potential customer population to lower sexual activity groups would imply a much lower average frequency of attendance to clubs: for example, if we include all individuals with 10+ sexual partners, the average attendance frequency would be once every year. We now report a more extensive explanation of this assumption in the Methods.

To evaluate the robustness of our results with respect to this assumption, we added a sensitivity analysis where we allowed any individual to attend a club in a given week, until club capacity is reached. After recalibration,

transmission of mpox in clubs was estimated to be much lower compared to the main analysis, with a mean reproduction number in clubs of 1.3 (against 2.7 in the main analysis) and a mean proportion of cases acquired in clubs of 2.5% (against 10% in the main analysis). The lower importance of club transmission explains why the model estimates a smaller reduction of transmissibility in clubs (mean 36% vs. 87% in the main analysis) and requires a higher reduction of transmissibility in the sexual contact network (mean 78% vs. 61% in the main analysis). A higher attack rate was estimated in lower sexual activity groups and a lower attack rate in the higher sexual activity groups, supporting the conclusion that population immunity in high-risk groups did not contribute significantly to the downturn of the epidemics. We now report the methods and results of the sensitivity analysis in the Supplementary Material (Section 4.4).

Finally, while it is beneficial that the authors conducted a sensitivity analysis examining the impact of assortative mixing by sexual activity, decades of research has provided substantial evidence of the non-random mixing of sexual partnerships - particularly in MSM - by categories such as age, ethnicity, etc. These factors could also interact with other important features of this simulation such as level of sexual activity, club attendance, etc. Randomly assigned sexual partners seems an extreme oversimplification of this process. The authors should provide additional justification for this decision, provide evidence this is not the case amongst MSM in Italy, or show evidence that it does not impact their conclusions.

We agree with the reviewer that there may be other important sources of assortativity in mixing among MSM and that these may interact with each other. Unfortunately, available data do not allow to evaluate such nuances in sexual mixing patterns and further studies on the subject are warranted. The prevalence of non-white ethnic groups is very low in Italy (8% of the population is non-Italian and <4% is non-European, according to the National Institute for Statistics); therefore, we can assume this covariate to play a marginal role. For this reason, we did not implement a sensitivity analysis with respect to ethnicity. To evaluate whether age assortativity may affect the conclusions of our study, we performed an additional sensitivity analysis where individuals have a higher probability to have a sexual partnership with other individuals of both similar age and similar sexual activity. After recalibration of model parameters with the same criteria as in the main analysis, results and conclusions remained qualitatively and quantitatively similar. We now report the methods and results of the new sensitivity analysis in the Supplementary Material (Section 4.2).

REVIEWERS' COMMENTS

Reviewer #1 (Remarks to the Author):

The authors have responded to all of my clarifying comments and have improved the manuscript in terms of readability and clarity. I especially appreciate the additional sensitivity analysis for club attendance criteria.

I do not have any reservations in accepting the paper as is.

Reviewer #1 (Remarks on code availability):

The authors have provided a zipfile containing the code for reproducibility. I have not run the code, but appreciate the level of detail in the README file and including all input and data files.

Reviewer #2 (Remarks to the Author):

The Authors have addressed my original comments well.

The new simulation code base is much more readable, and appears to reflect the concept of the model as described in the paper.

Reviewer #2 (Remarks on code availability):

This is a heavy duty computational model.

I have checked that it compiles and runs; and there are no obvious errors in the code itself that I can find.

However, I've lacked the compute resource to re-run the whole ABC pipeline to double check the outputs.

Reviewer #3 (Remarks to the Author):

The response and changes to the manuscript are satisfactory.